# STRUCTURED ALIGNMENT NETWORKS

## ABSTRACT

Many tasks in natural language processing involve comparing two sentences to compute some notion of relevance, entailment, or similarity. Typically this comparison is done either at the word level or at the sentence level, with no attempt to leverage the inherent structure of the sentence. When sentence structure *is* used for comparison, it is obtained during a non-differentiable pre-processing step, leading to propagation of errors. We introduce a model of *structured alignments* between sentences, showing how to compare two sentences by matching their latent structures. Using a structured attention mechanism, our model matches possible spans in the first sentence to possible spans in the second sentence, simultaneously discovering the tree structure of each sentence and performing a comparison, in a model that is fully differentiable and is trained only on the comparison objective. We evaluate this model on two sentence comparison tasks: the Stanford natural language inference dataset and the TREC-QA dataset. We find that comparing spans results in superior performance to comparing words individually, and that the learned trees are consistent with actual linguistic structures.

## 1 INTRODUCTION

There are many tasks in natural language processing that require comparing two sentences: natural language inference (Bowman et al., 2015; Nangia et al., 2017) and paraphrase detection (Wang et al., 2017b) are classification tasks over sentence pairs, and question answering often requires an alignment between a question and a passage of text that may contain the answer (Voorhees & Tice, 2000; Tan et al., 2016; Rajpurkar et al., 2016; Joshi et al., 2017).

Neural models for these tasks almost always perform comparisons between the two sentences either at the word level (Parikh et al., 2016), or at the sentence level (Bowman et al., 2015). Word-level comparisons ignore the inherent structure of the sentences being compared, at best relying on a recurrent neural network such as an LSTM (Hochreiter & Schmidhuber, 1997) to incorporate some amount of context from neighboring words into each word's representation. Sentence-level comparisons can incorporate the structure of each sentence individually (Bowman et al., 2016; Tai et al., 2015), but cannot easily compare substructures between the sentences, as these are all squashed into a single vector. Some models *do* incorporate sentence structure by comparing subtrees between the two sentences (Zhao et al., 2016; Chen et al., 2017), but require pipelined approaches where a parser is run in a non-differentiable preprocessing step, losing the benefits of end-to-end training.

In this paper we propose a method, which we call *structured alignment networks*, to perform comparisons between substructures in two sentences, without relying on an external, non-differentiable parser. We use a structured attention mechanism (Kim et al., 2017; Liu & Lapata, 2017b) to compute a *structured alignment* between the two sentences, jointly learning a latent tree structure for each sentence and aligning spans between the two sentences.

Our method constructs a CKY chart for each sentence using the inside-outside algorithm (Manning et al., 1999), which is fully differentiable (Li & Eisner, 2009; Gormley et al., 2015). This chart has a node for each possible span in the sentence, and a score for the likelihood of that span being a constituent in a parse of the sentence, marginalized over all possible parses. We take these two charts and find alignments between them, representing each span in each sentence with a structured attention over spans in the other sentence. These span representations, weighted by the span's likelihood, are then used to compare the two sentences. In this way we can perform comparisons between

sentences that leverage the internal structure of each sentence in an end-to-end, fully differentiable model, trained only on one final objective.

We evaluate this model on several sentence comparison datasets. In experiments on SNLI (Bowman et al., 2015) and TREC-QA (Voorhees & Tice, 2000), we find that comparing sentences at the span level consistently outperforms comparing at the word level. Additionally, and in contrast to prior work (Williams et al., 2017), we find that learning sentence structure on the comparison objective results in well-formed trees that closely mimic syntax. Our results provide strong motivation for incorporating latent structure into models that implicitly or expliclty compare two sentences.

## 2 WORD-LEVEL COMPARISON BASELINE

We first describe a common word-level comparison model, called *decomposable attention* (Parikh et al., 2016). This model was first proposed for the natural language inference task, but similar mechanisms have been used in many other places, such as for aligning question and passage words in the bi-directional attention model for question answering (Seo et al., 2016). This model serves as our main point of comparison, as our latent tree matching model simply replaces the word-level comparisons done in decomposable attention with span comparisons.

The decomposable attention model consists of three steps: *attend*, *compare*, and *aggregate*. As input, the model takes two sentences $a$ and $b$ represented by sequences of word embeddings $[\mathbf{a}_1, \cdots, \mathbf{a}_m]$ and $[\mathbf{b}_1, \cdots, \mathbf{b}_m]$. In the *attend* step, the model computes attention scores for each pair of tokens across the two input sentences and normalizes them as a soft alignment from $a$ to $b$ (and vice versa):

$$e_{ij} = F(\boldsymbol{a}_i)^T F(\boldsymbol{b}_j) \tag{1}$$

$$\boldsymbol{\beta}_i = \sum_{j=1}^{n} \frac{exp(\boldsymbol{e}_{ij})}{\sum_{k=1}^{n} exp(\boldsymbol{e}_{ik})} \boldsymbol{b}_j \tag{2}$$

$$\boldsymbol{\alpha}_i = \sum_{i=1}^{m} \frac{exp(\boldsymbol{e}_{ii})}{\sum_{k=1}^{m} exp(\boldsymbol{e}_{ki})} \boldsymbol{a}_i \tag{3}$$

where $F$ is a feed-forward neural network, $\boldsymbol{\beta}_i$ is the weighted summation of the tokens in $b$ that are softly aligned to token $a_i$ and vice versa for $\boldsymbol{\alpha}_i$.

In the *compare* step, the input vectors $\boldsymbol{a}_i$ and $\boldsymbol{b}_i$ are concatenated with their corresponding attended vector $\boldsymbol{\beta}_i$ and $\boldsymbol{\alpha}_i$, and fed into a feed-forward neural network, giving a comparison between each word and the words it aligns to in the other sentence:

$$\boldsymbol{v}_{ai} = G([\boldsymbol{a}_i, \boldsymbol{\beta_i}]) \quad \forall i \in [1, \cdots, m] \tag{4}$$

$$\boldsymbol{v}_{bj} = G([\boldsymbol{b}_j, \boldsymbol{\alpha_j}]) \quad \forall j \in [1, \cdots, n] \tag{5}$$

The *aggregate* step is a simple summation of $\boldsymbol{v}_{ai}$ and $\boldsymbol{v}_{bj}$ for each token in sentence $a$ and $b$, and the two resulting fixed-length vectors are concatenated and fed into a linear layer, followed by a softmax layer for classification:

$$\boldsymbol{v}_a = \sum_{i=1}^{m} \boldsymbol{v}_{ai} \tag{6}$$

$$\boldsymbol{v}_b = \sum_{j=1}^{n} \boldsymbol{v}_{nj} \tag{7}$$

$$\boldsymbol{y} = softmax(H([\boldsymbol{v}_a, \boldsymbol{v}_b])) \tag{8}$$

The decomposable attention model completely ignores the order and context of words in the sequence. There are some efforts strengthening decomposable attention model with a recurrent neural

A: Boeing is in Seattle

B: Boeing is a company based in WA

Figure 1: Example span alignments of a sentence pair, where different colors indicate matching spans. Note that some spans overlap, which cannot happen in a single tree; our model considers *all possible* span comparisons, weighted by the spans' marginal likelihood.

network (Liu & Lapata, 2017b) or intra-sentence attention (Parikh et al., 2016). However, these models amount to simply changing the input vectors $\boldsymbol{a}$ and $\boldsymbol{b}$, and still only perform a token-level alignment between the two sentences.

## 3 STRUCTURED ALIGNMENT NETWORKS

Language is inherently tree structured, and the meaning of sentences comes largely from composing the meanings of subtrees (Chomsky, 2002). It is natural, then, to compare the meaning of two sentences by comparing their substructures (MacCartney & Manning, 2009). For example, when determining the relationship between "Boeing is in Seattle" and "Boeing is a company based in WA", the ideal units of comparison are spans determined by subtrees: "in Seattle" compared to "based in WA", etc. (see Figure 1).

The challenge with comparing spans drawn from subtrees is that the tree structure of the sentence is latent and must be inferred, either during pre-processing or in the model itself. In this section we present a model that operates on the *latent* tree structure of each sentence, comparing *all possible* spans in one sentence with *all possible* spans in the second sentence, weighted by *how likely* each span is to appear as a constituent in a parse of the sentence. We use the non-terminal nodes of a binary constituency parse to represent spans. Because of this choice of representation, we can use the nodes in a CKY parsing chart to efficiently marginalize span likelihood over all possible parses for each sentence, and compare nodes in each sentence's chart to compare spans between the sentences.

### 3.1 LEARNING LATENT CONSTITUENCY TREES

A constituency parser can be partially formalized as a graphical model with the following cliques (Klein & Manning, 2004): the latent variables $c_{ijk}$ for all $i < j$, indicating the span from the $i$-th token to the $j$-th token ($span_{ij}$) is a constituency node built from the merging of sub-node $span_{ik}$ and $span_{(k+1)j}$. Given the sentence $x = [x_i, \cdots, x_n]$, the probability of a parse tree $z$ is,

$$p(c|x) = \frac{\exp(\sum_{c_{ijk} \in z} c_{ijk})}{\sum_{z \in Z} \exp(\sum_{c_{ijk} \in c} c_{ijk})} \tag{9}$$

where $Z$ represents all possible constituency trees for $x$.

The parameters to the graph-based CRF constituency parser are the unary potentials $\gamma_i$, reflecting the score of the token $x_i$ forming a unary constituency node and $\delta_{ikj}$ reflecting the score of $span_{ij}$ forming a binary constituency node with $k$ as the splitting point. It is possible to calculate the marginal probability of each constituency node $p(c_{ijk} = 1|x)$ using the inside-outside algorithm (Klein & Manning, 2003), and marginalize on the splitting points with $p(s_{ij} = 1|x) = \sum_{i \le k < j} p(c_{ijk} = 1|x)$ to compute the probability for a $span_{ij}$ being a constituency node. The inside-outside algorithm is constrained to generate a binary tree; this is not a severe limitation, however, as most structures can be easily binarized (Finkel et al., 2008).

In a typical constituency parser, the score $\delta_{ikj}$ is parameterized according to the production rules of a grammar, e.g., with normalized categorical distributions for each non-terminal. Our unlabeled grammar effectively has only a single production rule, however, so we instead parameterize these scores as multi-layer perceptrons operating on the representations of the subtrees being combined. For computational and statistical efficiency given this parameterization, we drop the dependence on the splitting point in this score, resulting in a score for each span $\delta_{ij}$ representing how

"constituent-like" the span is, independent of the merging of its children in the tree. This allows for a slightly-modified computation of the inside score in the inside-outside algorithm. Where the inside score $\alpha_{ij}$ is typically computed as $\alpha_{ij} = \sum_{i \leq k < j} \delta_{ikj} \alpha_{ik} \alpha_{(k+1)j}$, we instead compute it as $\alpha_{ij} = \delta_{ij} \sum_{i \leq k < j} \alpha_{ik} \alpha_{(k+1)j}$.

Up to this point, the tags of constituency nodes are not considered[1], leading to an unlabeled tree structure. However, with the binary tree constraint, not all tree nodes are syntactically complete, and thus some nodes may not be useful for comparison between the sentences. To overcome this, we introduce two artificial tags $T_0$ and $T_1$, where the former tag represents that this is a comparable constituent and the latter represents that this is just an intermediate node. In other words, the $T_1$ tag gives the model a fallback option when the span should not be compared to other spans, but is still helpful to building the tree structure. The inside pass is described in Algorithm 1, where $\gamma_i^0$ and $\gamma_i^1$ are unary potentials for the $i$-th word being a unary constituent with $T_0$ and $T_1$, and $\delta_i^0$ and $\delta_i^1$ are potentials for a span being a constituent with $T_0$ and $T_1$.

---

**Algorithm 1** The variant of the Inside algorithm

1: **for** $k$:=1 to $n$: **do**
2:     $\alpha_{kk}^0 = \gamma_k^0$
3:     $\alpha_{kk}^1 = \gamma_k^1$
4: **end for**
5: **for** $width$:=2 to $n$ **do**
6:     **for** $i$:=1 to $n - width + 1$ **do**
7:         $j := i + width - 1$
8:         **for** $k$:=$i + 1$ to $k$ **do**
9:             $\alpha_{ij}^0 += \delta_{ij}^0 (\alpha_{ik}^0 \alpha_{(k+1)j}^0 + \alpha_{ik}^1 \alpha_{(k+1)j}^1 + \alpha_{ik}^0 \alpha_{(k+1)j}^1 + \alpha_{ik}^1 \alpha_{(k+1)j}^0)$
10:            $\alpha_{ij}^1 += \delta_{ij}^1 (\alpha_{ik}^0 \alpha_{(k+1)j}^0 + \alpha_{ik}^1 \alpha_{(k+1)j}^1 + \alpha_{ik}^0 \alpha_{(k+1)j}^1 + \alpha_{ik}^1 \alpha_{(k+1)j}^0)$
11:        **end for**
12:    **end for**
13: **end for**
14: **return** $\alpha^0, \alpha^1$

---

The $\alpha$ values are the inside scores for all the spans in the sentence, which are basically the unnormalized scores indicating the whether the spans are proper constituents. After feeding these values into the outside algorithm, we can obtain the normalized marginal probability for each span $[\rho_{11}, \rho_{01}, \cdots, \rho_{ij}, \cdots, \rho_{(n-1)n}, \rho_{nn}]$, where $1 \leq j \leq n, 1 < i \leq j$ .

When computing the unary and binary potentials $\gamma$ and $\delta$, we use Long Short-Term Memory Neural Networks (LSTMs) (Hochreiter & Schmidhuber, 1997) and LSTM span features (Cross & Huang, 2016; Liu & Lapata, 2017a) for representing all the spans. We represent each sentence as a sequence of word embeddings $[\boldsymbol{w}_{sos}, \boldsymbol{w}_1, \cdots, \boldsymbol{w}_t, \cdots, \boldsymbol{w}_n, \boldsymbol{w}_{eos}]$. We run a bidirectional LSTM over the sentence and obtain the output vector sequence $[\boldsymbol{h}_0, \cdots, \boldsymbol{h}_t, \cdots, \boldsymbol{h}_{n+1}]$, where $\boldsymbol{h}_t = [\vec{\boldsymbol{h}}_t, \overleftarrow{\boldsymbol{h}}_t]$ is the output vector for the $t^{\text{th}}$ token, and $\vec{\boldsymbol{h}}_t$ and $\overleftarrow{\boldsymbol{h}}_t$ are the output vectors from the forward and backward directions, respectively. We represent a constituent $c$ from position $i$ to $j$ with a span vector $\boldsymbol{sp}_{ij}$ which is the concatenation of the vector differences $\vec{\boldsymbol{h}}_{j+1} - \vec{\boldsymbol{h}}_i$ and $\overleftarrow{\boldsymbol{h}}_{i-1} - \overleftarrow{\boldsymbol{h}}_j$:

And the potentials are computed by:

$$\gamma_{ij}^0 = MLP_u^0(\boldsymbol{w}_i), \ \gamma_{ij}^1 = MLP_u^1(\boldsymbol{w}_i) \tag{10}$$

$$\delta_{ij}^0 = MLP_b^0(\boldsymbol{sp}_{ij}), \ \delta_{ij}^1 = MLP_b^1(\boldsymbol{sp}_{ij}) \tag{11}$$

where $MLP_{T_0}$ and $MLP_{T_1}$ are two multilayer perceptions with a scalar output and ReLU as the activation function for the hidden layer.

---

[1] In computational linguistics, the tags are usually part-of-speech or phrase labels, such as *Proper noun, plural, Coordinating conjunction*, or *Noun phrase*.

After applying the parsing process on two sentences, we will get the marginal probability for all potential spans of the two constituency trees, which can then be used for aligning.

## 3.2 LEARNING STRUCTURED ALIGNMENTS

After learning latent constituency trees for each sentence, we are able to do span-level comparisons between the two sentences, instead of the word-level comparisons done by the decomposable attention model. The structure of these two comparison models are the same, but the basic elements of our structured alignment model are spans instead of words, and the marginal probabilities output from the inside-outside algorithm are used as a *re-normalization* value for incorporating structural information into the alignments.

For sentence $a$, with LSTM span features, we can obtain the representation for all potential spans, $[\boldsymbol{sp}_{11}^a, \boldsymbol{sp}_{12}^a, \cdots, \boldsymbol{sp}_{ij}^a, \cdots, \boldsymbol{sp}_{(m-1)m}^a, \boldsymbol{sp}_{mm}^a]$ and the marginal probability for them $[\rho_{11}^a, \rho_{12}^a, \cdots, \rho_{ij}^a, \cdots, \rho_{(m-1)m}^a, \rho_{mm}]$. And for sentence $b$, we can also get $[\boldsymbol{sp}_{11}^b, \cdots, \boldsymbol{sp}_{nn}^b]$ and $[\rho_{11}^b, \cdots, \rho_{nn}^b]$.

The attention scores are computed between all pairs of spans across the two sentences, and the attended vectors can be calculated as:

$$e_{ij,kl} = F(\boldsymbol{sp}_{ij}^a)^T F(\boldsymbol{sp}_{kl}^b) \tag{12}$$

$$\boldsymbol{\beta}_{ij} = \sum_{k=1}^n \sum_{l=k}^n \frac{exp(e_{ij,kl} + ln(\boldsymbol{\rho}_{kl}^b))}{\sum_{s=1}^n \sum_{t=s}^n (exp(e_{ij,st} + ln(\boldsymbol{\rho}_{st}^b))} \boldsymbol{sp}_{kl}^b \tag{13}$$

$$\boldsymbol{\alpha}_{kl} = \sum_{i=1}^m \sum_{j=i}^m \frac{exp(e_{ij,kl} + ln(\boldsymbol{\rho}_{ij}^a))}{\sum_{s=1}^m \sum_{t=s}^m exp(e_{st,kl} + ln(\boldsymbol{\rho}_{st}^a))} \boldsymbol{sp}_{ij}^a \tag{14}$$

here the method is similar to the process in the decomposable attention model, but the basic elements are text spans instead of tokens, and the marginal probabilities output from the inside-outside algorithm are used as a *re-normalization* value for incorporating structural information into the alignments.

Then, the span vectors are concatenated with the attended vectors and fed into a feed-forward neural network:

$$\boldsymbol{v}_{ij}^a = G([\boldsymbol{a}_{ij}, \boldsymbol{\beta}_{ij}]) \tag{15}$$

$$\boldsymbol{v}_{kl}^b = G([\boldsymbol{b}_{kl}, \boldsymbol{\alpha}_{kl}])\boldsymbol{\rho} \tag{16}$$

To aggregate these vectors, instead of using a direct summation, here we apply a weighted summation with the marginal probabilities as weights:

$$\boldsymbol{v}_a = \sum_{i=1}^m \sum_{j=i}^m \boldsymbol{\rho}_{ij}^a \boldsymbol{v}_{ij}^a \tag{17}$$

$$\boldsymbol{v}_b = \sum_{k=1}^n \sum_{l=1}^n \boldsymbol{\rho}_{kl}^b \boldsymbol{v}_{kl}^b \tag{18}$$

Here $\boldsymbol{\rho}^a$ and $\boldsymbol{\rho}^b$ work like the self-attention mechanism in (Lin et al., 2017) to replace the summation pooling step. The final output will still be obtained by a softmax function:

$$\boldsymbol{y} = softmax(H([\boldsymbol{v}_a, \boldsymbol{v}_b])) \tag{19}$$

## 4 EXPERIMENTS

We evaluate our structured alignment model with two natural language matching tasks: question answering as sentence selection and natural language inference. Since our approach can be considered as a module for replacing the widely-used token-level alignment, and can be plugged into other

neural models, the experiments are not intended to show that our approach can beat state-of-the-art baselines, but to test whether these methods can be trained effectively in an end-to-end fashion, can yield improvements over standard token-level alignment models, and can learn plausible latent constituency tree structures.

## 4.1 ANSWER SENTENCE SELECTION

We first study the effectiveness of our model for answer sentence selection tasks. Given a question, *answer sentence selection* is the task of ranking a list of candidate answer sentences based on their relatedness to the question, and the performance is measured by the mean average precision (MAP) and mean reciprocal rank (MRR). We experiment on the TREC-QA dataset (Wang et al., 2007), in which all questions with only positive or negative answers are removed. This leaves us with 1162 training questions, 65 development questions and 68 test questions. Experimental results of the state-of-the-art models and our structured alignment model are listed in Table 1, where the performances are evaluated with the standard TREC evaluation script.

The baseline model is the token-level decomposable attention strengthened with a bidirectional LSTM at the bottom for obtaining a contextualized representation for each token. For selecting the answer sentences, we consider this as a binary classification problem and the final ranking is based on the predicted possibility of being positive. We use 300-dimensional 840B GloVe word embeddings (Pennington et al., 2014) for initialization. The hidden size for BiLSTM is 150 and the feed-forward neural networks $F$ and $G$ are two-layer perceptrons with ReLU as activation function and 300 as hidden size. We apply dropout to the output of the BiLSTM and two-layer perceptrons with dropout ratio as 0.2. All parameters (including word embeddings) were updated with Adagrad (Duchi et al., 2011), and the learning rate was set to 0.05. Since the structure of the question and the answer sentence may be different, we use two variants of the structured alignment model in the experiment; the first shares parameters for computing the structures and the second uses separate parameters.

| Models | MAP | MRR |
|---|---|---|
| QA-LSTM (Tan et al., 2017) | 0.730 | 0.824 |
| Attentive Pooling Network (Santos et al., 2016) | 0.753 | 0.851 |
| Pairwise Word Interaction (He & Lin, 2016) | 0.777 | 0.836 |
| Lexical Decomposition and Composition (Wang et al., 2016) | 0.771 | 0.845 |
| Noise-Contrastive Estimation (Rao et al., 2016) | 0.801 | 0.877 |
| BiMPM (Wang et al., 2017b) | 0.802 | 0.875 |
| Decomposable Attention (Parikh et al., 2016) | 0.764 | 0.842 |
| Structured Alignment (Shared Parameters) (ours) | 0.770 | 0.850 |
| Structured Alignment (Separated Parameters) (ours) | 0.776 | 0.850 |

Table 1: Results of our models (bottom) and others (top) on the TREC-QA test set.

From the results we can see that on both the MAP and MRR metrics, structured alignment models perform better than the decomposable attention model, showing that the structural bias is helpful for matching the question to the correct answer sentence. Furthermore, the setting of separated parameters achieves higher scores on both metrics.

## 4.2 NATURAL LANGUAGE INFERENCE

The second task we consider is natural language inference, where the input is two sentences, a premise and a hypothesis, and the goal is to predict whether the premise entails the hypothesis, contradicts the hypothesis, or neither. For this task, we use the Stanford NLI dataset (Bowman et al., 2015). After removing sentences with unknown labels, we obtained 549,367 pairs for training, 9,842 for development and 9,824 for testing.

The baseline decomposable attention model is the same as in the question answering task. The hidden size of the LSTM was set to 150. We used 300-dimensional Glove 840B vectors to initialize the word embeddings. All parameters (including word embeddings) were updated with Adagrad (Duchi et al., 2011), and the learning rate was set to 0.05. The hidden size of the two-layer perceptrons was

| Models | Acc |
|---|---|
| Classifier with handcrafted features (Bowman et al., 2015) | 78.2 |
| LSTM encoders (Bowman et al., 2015) | 80.6 |
| Stack-Augmented Parser-Interpreter Neural Net (Bowman et al., 2016) | 83.2 |
| LSTM with inter-attention (Rocktäschel et al., 2016) | 83.5 |
| Matching LSTMs (Wang & Jiang, 2015) | 86.1 |
| LSTMN with deep attention fusion (Cheng et al., 2016) | 86.3 |
| Enhanced BiLSTM Inference Model (Chen et al., 2016) | **88.0** |
| Densely Interactive Inference Network Gong et al. (2017) | **88.0** |
| Decomposable Attention (Parikh et al., 2016) | 85.8 |
| Structured Alignment (ours) | 86.6 |

Table 2: Test accuracy on the SNLI dataset.

set to 300 and dropout was used with ratio 0.2. The structured alignment model in this experiment uses shared parameters for computing latent tree structures, since both the premise and hypothesis are declarative sentences.

The results of our experiments are shown in Table 2. Our structured alignment model gains almost a full point of accuracy (a 6% error reduction) over the baseline word-level comparison model with no additional annotation, simply from introducing a structural bias in the alignment between the sentences.

Table 2 shows the performances of the state-of-the-art models and our approaches. Similar to the answer selection task, the tree matching model outperforms the decomposable model stably.

### 4.3 ANALYSIS OF LEARNED TREE STRUCTURES

Here we give a brief qualitative analysis of the automatically learned tree structures. We present the CKY charts for two randomly-selected sentences in the SNLI test set in Figure 2. Recall that the CKY chart shows the likelihood of each span appearing as a constituent in the parse of the sentence, marginalized over all possible parses. By looking at these span probabilities, we can see that the model learned a model of sentence structure that corresponds well to known syntactic structures.

In the first example, we can see that "five children playing soccer" is a very likely span, as is "chase after a ball". Nonsensical spans, such as "playing soccer chase", have very low probability. In the second example, we can see that the model can even resolve some attachment ambiguities correctly. The prepositional phrase "at a large venue", which our model correctly identifies as a likely constituent in this sentence, has a very low score for attaching to "music" to form the constituent "music at a large venue". Instead, the model (correctly) prefers to attach "at a large venue" to "playing", giving the span "playing music at a large venue".

Our model is able to recover tree structures that very closely mimic syntax, without ever being given any access to syntactic supervision. This is in contrast to prior work by Williams et al. (2017), who were unable to learn syntax trees from a semantic objective. We use the same supervision as their model; we hypothesize that the difference in result is that they were trying to learn tree structures for each sentence *independently*, only performing comparisons at the sentence level. Comparing spans directly forces the model to induce trees with comparable constituents, giving the model a strong signal that was lacking in prior work.

## 5 RELATED WORK

**Sentence comparison models:** The Stanford natural language inference dataset (Bowman et al., 2015), and the expanded multi-genre natural language inference dataset (Nangia et al., 2017), are the most well-known recent sentence comparison tasks. The literature of models addressing this comparison task is far too extensive to include here, though the recent shared task on Multi-NLI gives a good survey of sentence-level comparison models (Nangia et al., 2017). Some of these sentence-level comparison models do use sentence structure, obtained either latently (Bowman et al.,

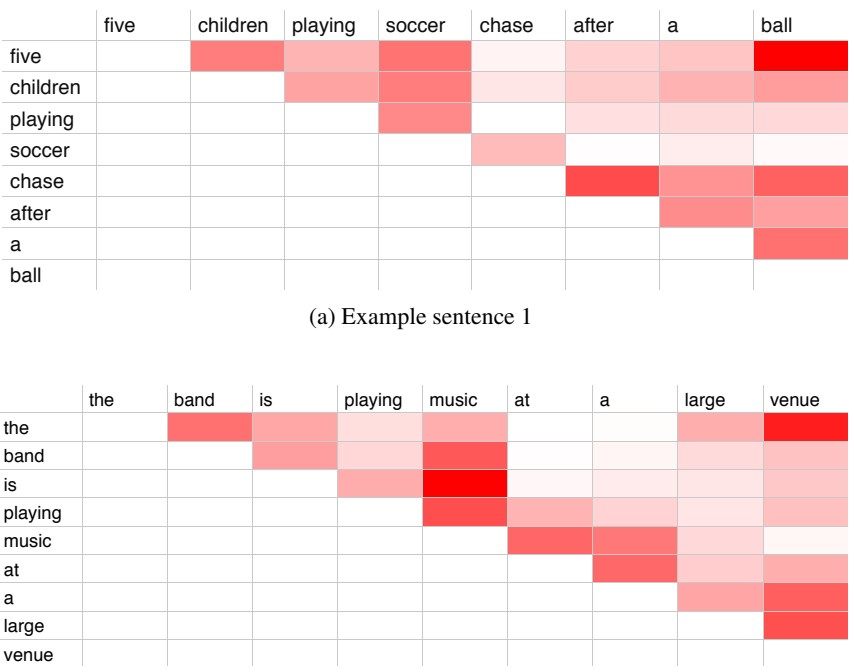

(a) Example sentence 1

(b) Example sentence 2

Figure 2: CKY charts showing marginalized span probabilities for two sentences in the SNLI test set. Each cell uses depth of the color to represent the probability of the span from the $i$-th word to the $j$-th word forming a proper constituent. Both trees capture the correct linguistic structure of the sentence.

2016) or during pre-processing (Zhao et al., 2016), but they squash all of the structure into a single vector, losing the ability to easily compare substructures between the two sentences.

For models doing a word-level comparison, the decomposable attention model, which we have discussed already in this paper (Parikh et al., 2016), is the most salient example, though many similar models exist in the literature (Chen et al., 2017; Wang et al., 2017b). The idea of word-level alignments between a question and a passage of text is also pervasive in the recent question answering literature (Seo et al., 2016; Wang et al., 2017a).

Finally, and most similar to our model, there have been many sentence comparison models proposed that directly compare subtrees between the two sentences (Chen et al., 2017; Zhao et al., 2016). However, all of these models are pipelined; they obtain the sentence structure in a non-differentiable preprocessing step, losing the benefits of end-to-end training. Ours is the first model to allow comparison between *latent* tree structures, trained end-to-end on the comparison objective.

**Structured attention:** While it has long been known that inference in graphical models is differentiable (Li & Eisner, 2009; Domke, 2011), and using inference in, e.g., a CRF (laf, 2001) as the last layer in a neural network is common practice (Liu & Lapata, 2017a; Lample et al., 2016), including inference algorithms as intermediate layers in end-to-end neural networks is a recent development. Kim et al. (2017) were the first to use inference to compute structured attentions over latent sentence variables, inducing tree structures trained on the end-to-end objective. Liu & Lapata (2017b) showed how to do this more efficiently, though their work was still limited to structured attention over a single sentence. Our model is the first to include latent structured alignments between two sentences.

**Inferring latent trees:** Unsupervised grammar induction is a well-studied problem (Cohen & Smith, 2009). The most recent work in this direction was the Neural E-DMV model of Jiang et al. (2016). While our goal is not to induce a grammar, we do produce a probabilistic grammar as a byproduct of our model. Our results suggest that training on more complex objectives may be a

good way to pursue grammar induction in the future; forcing the model to construct consistent, comparable subtrees between the two sentences is a strong signal for grammar induction.

# 6 CONCLUSION

We have considered the problem of comparing two sentences in natural language processing models. We have shown how to move beyond word- and sentence-level comparison to comparing spans between the two sentences, without the need for an external parser. Through experiments on several sentence comparison datasets, we have seen that span comparisons consistently outperform word-level comparisons, with no additional supervision. We additionally found our model was able to discover latent tree structures that closely mimic syntax, without any syntactic supervision.

Our results have several implications for future work. First, the success of span comparisons over word-level comparisons suggests that it may be profitable to include such comparisons in more complex models, either for comparing two sentences directly, or as intermediate parts of models for more complex tasks, such as reading comprehension. Second, though we have not yet done a formal comparison with prior work on grammar induction, our model's ability to infer trees that look like syntax from a semantic objective is intriguing, and suggestive of future opportunities in grammar induction research. Also, the speed of the model remains a problem, with the inside-outside algorithm involved, the speed of the full model will be 15-20 times slower than the decomposable attention model, mainly due the the fact this dynamic programming method can not be effectively accelerated on a GPU.

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
