# OpenReview forum: "STRUCTURED ALIGNMENT NETWORKS"
_ICLR.cc/2018/Conference — Reject_

### Official Review · AnonReviewer2 · 2017-11-27

**Rating:** 6
**Confidence:** 4

**Review:**

Summary:
This paper introduces a structured attention mechanisms to compute alignment scores among all possible spans in two given sentences. The span representations are weighted by the spans marginal scores given by the inside-outside algorithm. Experiments on TREC-QA and SNLI show modest improvement over the word-based structured attention baseline (Parikh et al., 2016).

Strengths:
The idea of using latent syntactic structure, and computing cross-sentence alignment over spans is very interesting.

Weaknesses:
The paper is 8.5 pages long.

The method did not out-perform other very related structured attention methods (86.8, Kim et al., 2017, 86.9, Liu and Lapata, 2017)

Aside from the time complexity from the inside-outside algorithm (as mentioned by the authors in conclusion), the comparison among all pairs of spans is O(n^4), which is more expensive. Am I missing something about the algorithm?

It would be nice to show, quantitatively, the agreement between the latent trees and gold/supervised syntax. The paper claimed “the model is able to recover tree structures that very closely mimic syntax”, but it’s hard to draw this conclusion from the two examples in Figure 2.

---

### Official Review · AnonReviewer1 · 2017-11-28
**Structured alignment networks**

**Rating:** 5
**Confidence:** 4

**Review:**

This paper proposes a model of "structured alignments" between sentences as a means of comparing two sentences by matching their latent structures. Overall, this paper seems a straightforward application of the model first proposed by Kim et al. 2017 with latent tree attention.

In section 3.1, the formula for p(c|x) looks wrong: c_{ijk} are indicator variables. but where are the scores for each span? I think it should be c_{ijk} * \delta_{ijk} under the summations instead.

In the same section, the expression for \alpha_{ij} seems to assume that \delta_{ijk} = \dlta_{ij} regardless of k. I.e. there are no production rule scores (transitions). This seems rather limiting, can you comment on that?

In the answer selection and NLI experiments, the proposed model does not beat the SOTA, and is only marginally better than unstructured decomposable attention. This is rather disappointing.

The plots in Fig 2 with the marginals on CKY charts are not very enlightening. How do this marginals help solving the NLI task?

Minor comments:
- Sec. 3: "Language is inherently tree structured" -- this is debatable...
- page 8: (laf, 2008): bad formatted reference

---

### Official Review · AnonReviewer3 · 2017-12-04
**An intriguing idea, a few weaknesses however.**

**Rating:** 5
**Confidence:** 4

**Review:**

This paper describes the use of latent context-free derivations, using
a CRF-style neural model, as a latent level of representation in neural
attention models that consider pairs of sentences. The model implicitly
learns a distribution over derivations, and uses marginals under this
distribution to bias attention distributions over spans in one sentence
given a span in another sentence.

This is an intriguing idea. I had a couple of reservations however:

* The empirical improvements from the method seem pretty marginal, to the
point that it's difficult to know what is really helping the model. I would
liked to have seen more explanation of what the model has learned, and
more comparisons to other baselines that make use of attention over spans.
For example, what happens if every span is considered as an independent random
variable, with no use of a tree structure or the CKY chart?

* The use of the \alpha^0 vs. \alpha^1 variables is not entirely clear. Once they
have been calculated in Algorithm 1, how are they used? Do the \rho values
somewhere treat these two quantities differently?

* I'm skeptical of the type of qualitative analysis in section 4.3, unfortunately.
I think something much more extensive would be interesting here. As one
example, the PP attachment example with "at a large venue" is highly suspect;
there's a 50/50 chance that any attachment like this will be correct, there's
absolutely no way of knowing if the model is doing something interesting/correct
or performing at a chance level, given a single example.

---

### Comment · Area_Chair · 2017-12-27
**Rebuttal?**

Authors, please post a rebuttal soon if you are planning on it.

---

### Decision · Program_Chairs · 2018-01-29
**ICLR 2018 Conference Acceptance Decision**

**Decision:**

Reject

**Comment:**

This work introduces a new type of structured attention network that learn latent structured alignments between sentences in a fully differentiable manner, which allows the network to learn not only the target task, but also the latent relationships. Reviewers seem partial to the idea of the work, and it's originality, but have issues with the contributions. In particular:

- The reviewers note that the gains in performance from using this approach are quite small and do not outperform previous structured approaches.